# Comparative Effects of Stovers of Four Varieties of Common Vetch on Growth Performance, Ruminal Fermentation, and Nutrient Digestibility of Growing Lambs

**DOI:** 10.3390/ani10040596

**Published:** 2020-04-01

**Authors:** Yafeng Huang, Cory Matthew, Fei Li, Zhibiao Nan

**Affiliations:** 1State Key Laboratory of Grassland Agro-ecosystems, College of Pastoral Agriculture Science and Technology, Lanzhou University, Lanzhou 730020, China; huangyafeng316@163.com (Y.H.); lfei@lzu.edu.cn (F.L.); 2School of Agriculture and Environment, Massey University, Private Bag 11 222, Palmerston North 4442, New Zealand; C.Matthew@massey.ac.nz

**Keywords:** common vetch, nutritive quality, lamb growth, volatile fatty acid, methane emission

## Abstract

**Simple Summary:**

Common vetch is an important legume crop of mixed crop-livestock systems, and it has the ability to satisfy food, fodder, and fertilizer demands through grain, stover, and nitrogen fixation. The aim of this study was to evaluate common vetch varietal differences in stover nutritive value, ruminal fermentation properties, nutrient digestibility, nitrogen retention, and animal performance in fattening lambs consuming a diet comprising 20% common vetch. The results showed that the substitution of alfalfa hay by varieties Lanjian No. 1 and Lanjian No. 2 stovers in ruminant diets could be advantageous in reducing dependence on imported alfalfa hay, thereby enhancing sustainability of small holder farmers on the Tibetan Plateau.

**Abstract:**

This study evaluated common vetch stover as a feed in mixed rations for growing lambs. Four common vetch varieties were compared with alfalfa (control) for their effects on growth performance, ruminal fermentation, nutrient digestibility, and nitrogen retention. Male Hu lambs (*n* = 50) aged 3 months, with a mean body weight of 17.5 ± 0.34 kg were allocated randomly to one of the five dietary treatments, making 10 lambs per treatment. The experiment lasted 67 days with a 10-day adaptation period and a 50-day fattening period, and with the final 7 days used for a nutrient digestibility and nitrogen balance trial. All diets contained 30.0% maize straw and 50.0% concentrate, with different forage sources (on a fed basis): 20.0% alfalfa hay (control), 20.0% local common vetch variety 333A (C333A) stover, or 20.0% stover of one of three improved common vetch varieties: Lanjian No. 1 (CLJ1), Lanjian No. 2 (CLJ2), or Lanjian No. 3 (CLJ3). For stover quality, CLJ1 stover had the greatest crude protein (CP), in vitro organic matter digestibility (IVOMD), and metabolizable energy (ME) content and the least cell wall contents, while C333A stover had the least CP, IVOMD, and ME contents and the greatest cell wall contents. Sheep fed the control diet had a greater average daily gain (ADG), apparent digestibility of organic matter (DOM), neutral detergent fiber, acid detergent fiber, and nitrogen retention, and greater ruminal total volatile fatty acids concentration than lambs fed the C333A or CLJ3 diet, but similar performance to lambs fed the CLJ1 and CLJ2 diets. The feed conversion ratio and predicted CH_4_ emission per unit of DOM intake and ADG of the control, CLJ1, and CLJ2 diets was significantly lower (*p* < 0.05) than for the other diets. Based on these results, stovers of varieties CLJ1 and CLJ2 can be recommended as an alternative to alfalfa hay and for use in a legume crop rotation with cereals on the Tibetan plateau.

## 1. Introduction

The use of common vetch (*Vicia sativa* L.) has attracted increasing interest in smallholder crop-livestock systems, partly as an animal feed, utilizing the seed and stover and partly to maintain soil fertility as a nitrogen-fixing strategy [1,2], but also because of its suitability for cultivation in conditions where other feed legumes are poorly suited, including in regions with harsh winters (i.e., cold, dry conditions) and in alpine conditions [3,4]. Several reports show that crop residue after harvesting common vetch was clearly greater than the seed yield obtained (for example, 4592 kg/ha crop residue after harvesting 1436 kg/ha of seed), indicating a substantial potential for feeding of ruminant livestock on the stover [1,5]. Common vetch stover was reported by Larbi et al. [1] and Makkar et al. [6] to contain 12.7% crude protein (CP), and have 51.7% in vitro organic matter digestibility (IVOMD), respectively. These values are high enough for common vetch to be used as a source of CP supplement or replacement for ruminant animals offered low CP cereal straw or native pasture hay-based diets, as occurs in many smallholder crop-livestock systems. However, the use of common vetch stover as a livestock feed has historically been low, because of a lack of knowledge of the potential of crop residues for use in the feeding and production of ruminants, since a majority of the previous studies have focused on yield, chemical composition, and in vitro digestibility of stover [1,4,6].

The livestock sector has an important socioeconomic role in many countries and has rapidly expanded in several regions of the world to meet increased consumption demand for red meats [7]. In particular, the global demands for sheep meat are predicted to reach a rate of 21% per year from 2016 to 2026 [8]. In China, the country with the largest sheep meat production, mutton production is mainly based in the western and northwestern provinces, especially the Tibetan plateau. The Tibetan plateau is the headwater region of most of Asia’s major rivers and provides a livelihood for over 9.8 million pastoralists, despite having a fragile ecosystem [9]. To address the eco-environmental problems of supporting an expanding population on the Tibetan Plateau, an ecosystem conservation and restoration project was initiated in 2000, including the encouragement of pen-raising of sheep [10]. However, due to the long cold winter and short growing season, generating an adequate forage supply is a key limitation facing small holder farmers on the Tibetan Plateau [4,9].

Methane (CH_4_) emission by ruminants is a significant contributing factor to the emerging global warming phenomenon [11,12,13]. In addition, enteric CH_4_ formed by the fermented feed in the gastrointestinal tract of ruminant represents a loss of approximately 2–12% of gross energy (GE) intake [11]. Hassanat et al. [14] indicated that CH_4_ emission varied with the composition of forages in ruminant rations. Therefore, in areas of high livestock concentration, optimization of ration formulation to reduce CH_4_ emissions from ruminants is important for bringing enormous economic benefits and protection of the global environment.

Despite common vetch stover having potential as a feed resource for ruminant animals, no studies have been conducted to our knowledge on the effect of using common vetch stover in ruminant diets on growth performance. In addition, there is a need for information comparing the CH_4_ emission of animals fed common vetch stover with emissions of animals fed a more commonly used forage such as alfalfa (*Medicago Sativa* L.) hay. Accordingly, the objective of this study was to compare animal growth performance, ruminal fermentation, nutrient digestibility, and nitrogen (N) retention in fattening lambs fed diets containing a 20% addition of alfalfa hay or stover of one of four different varieties of common vetch.

## 2. Materials and Methods

### 2.1. Study Area, Common Vetch Stover Production, and Sampling

The study was carried out at Lanzhou University Xiahe experimental farm in Gansu, China (35°19′ N, 102°58′ E; 2880 m above sea level), located in the eastern margin of the Tibetan Plateau. The climate is continental, modified by high altitude, with an average annual rainfall of 448 mm (80% of this occurring between May and September) and a mean annual air temperature of 3.92 °C (average values for 2000 to 2016).

Plots of three improved common vetch varieties Lanjian No. 1 (CLJ1), Lanjian No. 2 (CLJ2), and Lanjian No. 3 (CLJ3), and of one local variety 333A (C333A) were planted and received common management during the main rainy season from May to September 2017. Seeds were sown by hand with 20 cm row spacing and 1.50 million viable seeds per ha. Before sowing, seeds were inoculated with rhizobium (CCBAU01069, China Agricultural University, Beijing, China), which was recommended based on the symbiont performance for these varieties [15]. Neither irrigation nor fertilizer was applied after sowing, and weeds were controlled by hand. For the fattening experiment, all plants at pod maturity were harvested by hand, air-dried in the field, and threshed for stover production for the feeding experiment. For stover quality analysis, eight plots (1 m × 1 m each) for each variety were randomly harvested under a similar drying method.

### 2.2. Animals, Experimental Design, and Growth Performance

Animals were maintained according to the guidelines set by the Biological Studies Animal Care and Use Committee of Gansu Province, China (2005–12). Fifty male Hu lambs with an average age of three months and an average initial body weight (BW) of 17.5 ± 0.34 kg were randomly allocated to one of five treatment groups, each with ten lambs. All diets contained 30% corn (*Zea mays* L.) straw, 50% concentrate mixtures, and 20% common vetch stover or 20% alfalfa hay as the control (Table 1). The diets varied in the variety of common vetch stover added, with the treatments being the alfalfa control as above, or 20% stover of one of the common vetch varieties C333A, CLJ1, CLJ2, or CLJ3. The dried forage and concentrate components were combined as pelleted total mixed rations (TMR). The diets were formulated according to feeding standards for meat-producing sheep and goats [16]. Ingredients and chemical composition of the experimental diets are shown in Table 1.

The lambs were treated for internal parasites before commencement of the study. The experimental period lasted 67 days, with an adaptation period of 10 days, a 50-day fattening period, and a 7-day digestibility trial. Throughout the experimental period, all lambs were randomly assigned to individual pens (3.1 m × 1.0 m × 0.85 m) equipped with feeders and water buckets. The daily feed ration was offered ad libitum to the lambs in two parts at 08:30 and 16:30 h. During the fattening period, amounts of feed offered were recorded daily and the orts were weighed weekly, for each lamb individually, to estimate dry matter intake (DMI). A two-day-average BW of lambs, with weighing carried out before the morning feeding, was recorded individually at the beginning and at the end of the fattening period to determine average daily gain (ADG) and feed conversion ratio (FCR = DMI/ADG).

### 2.3. Ruminal Fermentation Parameters

At the end of the fattening period, ruminal fluids were obtained 2–4 h after the morning feeding, from five lambs randomly selected from each treatment, using an orogastric tube [17] Ruminal fluid pH was measured immediately after sampling using a portable pH meter (PHB-4, Shanghai, China). Ruminal fluid samples were strained through four layers of cheesecloth, and samples of 10 mL of filtrate were preserved with 1 mL of 25% metaphosphoric acid, stored at –20 °C for volatile fatty acids (VFA) determination [18]. After thawing, samples of ruminal fluid were centrifuged (15,000× *g* for 15 min, at 4 °C) and then supernatant fluid was used to determine the VFA by a gas chromatography (GC522, Wufeng Instruments, Shanghai, China) [18].

### 2.4. Metabolism Trials

In the last 7 days of the study, five lambs per treatment were randomly selected and moved individually into metabolic crates equipped with feeders and water buckets to evaluate nutrient digestibility and nitrogen balance. The digestibility trial lasted for 7 days with a four-day adaptation period, which was followed by the total collection of feces and urine for three consecutive days. Daily fecal samples for each lamb were collected, weighed, mixed thoroughly, and a 20% subsample dried at 65 °C for 48 h. At the end of collection period, dried fecal samples were composited, and then 10% was stored at −4 °C for later analyses. Urine of each lamb was collected daily in a plastic container with 50 mL of 6 N HCl added to prevent ammonia nitrogen losses. Then, 5% of the collected urine each day from each lamb was deposited in a plastic container and stored at −20 °C for later nitrogen analysis. After the digestibility trial, samples of diets offered, orts, and feces were taken to estimate nutrient digestibility. Nitrogen retention was calculated as nitrogen intake minus the sum of fecal nitrogen + urinary nitrogen.

### 2.5. Laboratory Analyses

Samples of stover, diets offered, orts, and feces were oven-dried at 65 °C for 48 h and ground through a 1 mm sieve for further analysis. Chemical composition of the samples was analyzed using the standard methods of the Association of Official Analytical Chemists [19] for dry matter (DM; 135 °C in a forced-air oven for 2 h; method 930.15), ash (550 °C in an ashing furnace for 6 h; method 938.08), nitrogen (N; the Kjeldahl procedure; method 988.05), ether extract (EE; method 920.85), acid detergent fiber (ADF; method 973.18), acid detergent lignin (ADL; method 973.18), calcium (Ca; method 945.03), and phosphorus (P; method 965.17). The CP content was calculated by multiplying the N value by 6.25. The neutral detergent fiber (NDF) content was estimated with the addition of heat stable α-amylase and sodium sulfite according to the procedure of Van Soest et al. [20]. The NDF and ADF contents were expressed as the inclusive of residual ash. The IVOMD and metabolizable energy (ME) values of stover samples were calculated from in vitro gas production (GP) after 24 h of incubation [21]. Rumen fluid was obtained before the morning feeding from three adult fistulated Dorper × thin-tailed Han crossbred rams and mixed. These animals were fed Chinese wild rye (*Leymus chinensis* (Trin.) Tzvel.) hay and concentrates in a ratio of 1.5:1 with free access to water and mineral/vitamin licks. Briefly, approximately 200 mg of each sample (in triplicate) was incubated with 30 mL of rumen fluid–buffer mixture into a 100 mL graduated glass syringe. Each syringe was placed vertically in a water bath at 39 °C with three syringes without a sample used as blanks. The volume of GP was recorded after 24 h of incubation, corrected for the result from blanks, and employed to determine IVOMD (g/kg DM) and ME (MJ/kg DM). The syringes were held vertically at eye level and a gas reading of each syringe was manually recorded. The IVDMD and ME were estimated using the models of Menke and Steingass [21], as IVOMD = 148.8 + 8.89 GP + 0.45 CP + 0.0651 ASH; ME = 2.20 + 0.136 GP + 0.0057 CP + 0.000029 EE ^2^, where GP is the net gas production after 24 h of incubation (mL/200 mg DM), ASH is the ash content (g/kg DM), CP is the crude protein content (g/kg DM), and EE is the ether extract content (g/kg DM). The incubation run was repeated three times on different days.

### 2.6. Calculations and Statistical Analysis

Predicted methane emission was calculated stoichiometrically from ruminal VFA, according to Moss et al. [22] as: CH_4_ (mmol/L) = 0.45 AC+ 0.40 BC −0.275 PC, where AC, BC, and PC are acetate, butyrate, and propionate, respectively. Total CH_4_ emission (L/Day) was also estimated from the recorded DMI throughout the fattening period, following the prediction equation of Ramin and Huhtanen [23]: CH_4_ (L/Day) = 20 ( ± 12.1) + 35.8 ( ± 2.87) × DMI–0.50 ( ± 0.132) × DMI ^2^ (negative values were used to estimate the minimum value and positive values were used to estimate the maximum value for methane emission). Methane measurements were calculated based on the ADG and digestible organic matter intake (DOMI). For all response variables, analysis of variance was performed using a general linear model routine in SPSS software (Version 21.0 IBM Corporation, Armonk, NY, USA). Means were then separated using Duncan’s multiple range test with a significance threshold of *p* < 0.05.

## 3. Results

### 3.1. Stover Chemical Composition and in Vitro Characteristics

Significant differences (*p* < 0.01) among the varieties were observed for stover chemical composition (Table 2), with a general pattern of negative correlation between CP and the various fiber determinations, and positive correlation between CP and ME and OMD. Crude protein content varied from 117.62 to 142.72 g/kg DM, and was the highest in CLJ1 stover. The EE content of both C333A stover and CLJ1 stover was clearly lower (*p* < 0.001) than that of other varieties. The highest contents of OM, NDF, ADF, ADL, and cellulose were observed in C333A stover while CLJ1 stover had the least. Hemicellulose content ranged from 143.01 to 176.08 g/kg DM, with the least value corresponding to CLJ1 stover and the highest to CLJ3 stover. By contrast, P and Ca contents were highest in CLJ1 stover and least in C333A stover (*p* < 0.001). There were significant differences (*p* < 0.01) between stover of the different varieties in the IVOMD and ME. The IVOMD and ME values were least in C333A stover and highest in CLJ1 stover. 

### 3.2. Animal Growth Performance, Nutrient Digestibility, and Nitrogen Balance

As shown in Table 3, neither the initial nor the final BW of the lambs was affected by the dietary treatments (*p* > 0.05). The ADG by lambs fed the control diet was greater than that of lambs fed the C333A and CLJ3 diets (*p* = 0.010), but was not different from that of other diets. Despite DMI being similar among dietary treatments (*p* > 0.05), the FCR value was significantly greater in C333A and CLJ3 diets compared to the control, CLJ1, and CLJ2 diets (*p* < 0.001). 

Apparent DM digestibility (621.60 g/kg on average) was not affected by dietary treatments (*p* > 0.05). Apparent digestibility values of OM, NDF, and ADF were significantly lower (*p* < 0.05) in the C333A and CLJ3 diets than in the control diet, while no significant difference was observed between the CLJ1, CLJ2, and control diets. The apparent CP digestibility did not differ among the dietary treatments (*p* > 0.05), but there was a tendency to lower apparent CP digestibility in the C333A diet compared to the control diet (*p* = 0.092).

No effects of the dietary treatments were observed in the nitrogen intake and excretion of N in urine (*p* > 0.05; Table 3). Excretion of N in feces was least in lambs fed the control diet and highest in the C333A diet (*p* < 0.05), with lambs receiving the CLJ2 diet having intermediate values. Nitrogen retention significantly differed between dietary treatments (*p* < 0.05), with the least nitrogen retained observed for the C333A diet and the highest for the control diet. For N retention expressed as g/100 g N intake, lambs fed the control diet had a greater value than those fed the C333A, and CLJ3 diets, while values for those fed other diets did not differ.

### 3.3. Rumen Fermentation Characteristics

Ruminal pH averaged 6.65 and did not differ between the diets (*p* > 0.05; Table 4). Lambs fed the control diet had lower total VFA content in the ruminal fluid (*p* < 0.05) than lambs fed the C333A or CLJ3 diets but similar VFA content to lambs fed other diets. Regarding individual VFA, the molar proportions of butyrate, isobutyrate, and isovalerate were similar among dietary treatments, averaging 14.89%, 1.31%, and 1.65% of total VFA, respectively. The control diet resulted in a lower molar proportion of propionate than the C333A and CLJ3 diets, while ruminal propionate in lambs fed the other diets did not differ from the control diet. The molar proportions of acetate and valerate in ruminal fluid of lambs offered control, CLJ1, and CLJ2 diets was considerably greater (*p* < 0.01) than that of lambs fed the C333A and CLJ3 diets. The ratio of acetic to propionic acid ranged from 1.89 to 2.65, the lowest value corresponding to the C333A diet and the highest to the control diet (*p* < 0.01).

### 3.4. CH_4_ Emission

Table 5 presents results for predicting CH_4_ emission based on the models of Moss et al. [22] and Ramin and Huhtanen [23]. Regarding VFA production, mean CH_4_ emission from lambs offered the control, CLJ1, and CLJ2 diets was clearly greater than that from lambs offered the C333A or CLJ3 diets. Based on the day, mean CH_4_ production did not differ among dietary treatments, but for both DOM and ADG, predicted CH_4_ emission values were significantly greater for both the C333A and CLJ3 diets than for the control, CLJ1, and CLJ2 diets.

## 4. Discussion

This research is novel because, to our knowledge, it is the first study providing varietal comparison data on nutrient digestibility, rumen fermentation, predicted CH_4_ emission, and performance of growing lambs fed common vetch stover.

### 4.1. Stover Chemical Composition and in vitro Characteristics

Forage quality attributes are of high importance in evaluation of the animal performance production potential of a forage crop [1]. Significant differences in stover quality appear to relate to varietal variations in accumulation and lignification of cell wall contents, and the leaf to stem ratio. Others have reported similar results in common vetch [1,24] and faba bean (*Vicia faba* L.) [25]. With the exception of variety 333A, stover CP and IVOMD of all improved varieties exceeded 120 g/kg DM and 517 g/kg OM, respectively. These values are sufficient for stovers of the improved varieties to be employed as a source of CP supplements for ruminants offered low CP cereal straw or native pasture hay based diets [1], as commonly occurs in farm systems like those on the Tibetan plateau. Common vetch stover CP (75–134 g/kg DM) and OM ranges (852–927 g/kg DM) observed by Fortina et al. [26] are similar to our results, despite the greater NDF, ADF, and ADL ranges (555–774, 396–571, and 77–123 g/kg DM, respectively) compared to the present study. Larbi et al. [1] reported similar NDF and ADF ranges (378–505 and 324–353g/kg DM, respectively), and lower CP and IVOMD ranges (65–124 g/kg DM and 437–511 g/kg OM, respectively) compared to the current study. The stover NDF, ADF, and ME ranges obtained in this study fell within ranges reported by Makkar et al. [6]. However, our CP values were greater and the IVOMD levels were lower than those reported by Makkar et al. [6]. The differences in stover quality between studies, appear to the differences between varieties tested for stem:leaf ratio, crop maturity stage at harvest, post-harvest handling practice, and soil and climate conditions. Of the varieties tested, CLJ1 has the best potential for use as a ruminant feed, in terms of stover quality.

### 4.2. Animal Growth Performance, Nutrient Digestibility, and Nitrogen Balance

The ADG of Hu lambs aged about 90 days varied from 200.00 to 251.60 g/day, and these values were similar to those previously reported by Chai et al. [27]. The greater ADG of lambs offered control, CLJ1, and CLJ2 diets resulted in a greater FCR compared to other diets, despite the DMI not differing among treatments. Variable responses of FCR among the dietary treatments could be partly explained by greater digestibility of nutrients in the control, CLJ1, and CLJ2 diets. Haddad and Husein [28] reported that the BW gain of lambs offered an alfalfa-hay-based diet was greater than that of animals fed a diet incorporating stover of the vetch species *Vicia ervilia*, but similar to that of lambs offered a lentil-stover-based diet. 

The lambs fed the control and CLJ1 diets had greater apparent digestibility of OM, NDF, and ADF compared to those fed diets with C333A and CLJ3, despite there being no differences found for apparent digestibility of DM and CP between the diets. In this study, concentrate ingredients were similar among the experimental TMR diets; thus, the differences were attributed to their common vetch stovers/alfalfa hay traits reflected in differences in morphological (e.g., stem/leaf ratio) and chemical composition. Variable responses of apparent digestibility among the dietary treatments could be partly due to greater CP and lower ADL contents of these stovers, as these factors have a positive influence on the extent of digestion [29] These results are partially consistent with earlier studies on other crop residues on sorghum (*Sorghum bicolor* [L.] Moench) [30] and faba bean [31].

Reducing nitrogen excretion from animals to the environment has received great attention in mixed crop-livestock systems, as excess nitrate-nitrogen accumulation in the environment has a number of detrimental effects [32]. In this study, nitrogen excretion in urine was not affected by the dietary treatments. However, fecal N excretion of animals fed the control diet was lower than for animals fed the C333A or CLJ3 stover diets, while the values for other diets did no differ from those of the control diet. Variation among diets in nitrogen excretion in feces is likely explained by the lower apparent CP digestibility in the C333A and CLJ3 stover diets. This result is consistent with results of Corea et al. [33] who found that a greater apparent CP digestibility in cowpea (*Vigna sinensis*) hay included in diets, led to lower feed fecal nitrogen excretion efficiency. Nitrogen retention of animals depends on the amount of nitrogen excretion in urine and feces [30]. In the current study, nitrogen retention (both, in g/day and g/100 g N intake) was highest for the control diet and least for the C333A stover diet. This result suggests that feeding control, and CLJ1 and CLJ2 stover diets resulted in greater nitrogen utilization at least partly because of improved forage quality and improved digestibility. Based on these results for animal growth performance and nutrient digestibility, stovers of common vetch varieties CLJ1 and CLJ2 could be considered as an alternative to alfalfa hay.

### 4.3. Rumen Fermentation Characteristics

Ruminal pH is an important indicator of the status of the ruminal microbial ecosystem in ruminants [34]. The pH of the rumen liquid sampled in this study varied from 6.51 to 6.78, these values being within the normal physiological range of 6.5–6.7 as outlined by Van Soest [29]. Ruminal VFA is the main energy source for ruminants. Ruminal total VFA content of lambs fed the control diet was clearly greater than the ruminal VFA content of lambs fed the C333A and CLJ3 diets, thus indicating increased energy supply to support body weight gain from the control diet. Variable responses of ruminal VFA among the dietary treatments reflected differences between the diets tested in apparent digestibility of nutrients. The VFA results in this study were in agreement with the study of Willms et al. [35] who reported that the increases in apparent OM digestion, increased total VFA concentration. Wang et al. [36] also reported that greater total concentration of ruminal VFA in diets of feeding ruminants was indicative of a greater nutrient digestibility. In contrast, Karamnejad et al. [37] found that total VFA concentration and molar proportions of VFA were not influenced by the apparent digestibility of DM, OM, CP, and ADF. The inconsistency between the studies may be associated with differences in plant species, diet ingredient, and animal types.

### 4.4. Predicted CH_4_ Emission

Improvement of forage efficiency and reduction of CH_4_ emissions is of high importance to benefit the global eco-environment [12,13]. Based on the model of Moss et al. [22], the C333A and CLJ3 stover diets would generate lower CH_4_ emissions than the control, and CLJ1 and CLJ2 stover diets, due to differences in total VFA concentration and proportions of individual VFA. The lambs fed the C333A or the CLJ3 stover diets had greater predicted CH_4_ emission values based on the levels of apparent OM digestibility [23]. This result arose from the lower value for apparent OM digestibility for the lambs offered the C333A and CLJ3 stover diets, as no differences were found for CH_4_ emission values based on the day. In addition, Eckard et al. [38] and Soltan et al. [39] stated that in smallholder crop-livestock systems, the most important consideration is reducing CH_4_ emission per unit of milk production or body weight gain, rather than reduction of CH_4_ emission per unit of DMI, or relative to apparent OM digestibility. In this respect, the control, and CLJ1 and CLJ2 stover diets are identified in this study as providing the desired lower CH_4_ production per unit of BW gain. Therefore, stovers of common vetch varieties CLJ1 and CLJ2 have better potential for use as an alternative to alfalfa hay.

## 5. Conclusions

Evaluation of four common vetch varieties showed that CLJ1 stover had the highest CP, IVOMD, and ME contents, and the least cell wall content, compared to other varieties, with the exception of the IVOMD, ME, and ADL contents from CLJ2 stover. Sheep offered the control diet had greater ADG and nutrient digestibility, and lower predicted CH_4_ emission expressed relative to DOMI or ADG than lambs offered C333A and CLJ3 diets, but their performance did not differ from lambs offered CLJ1 or CLJ2 stover diets. Considering stover quality, animal performance, and methane emission, CLJ1 and CLJ2 stovers are identified in this study as viable alternatives to alfalfa hay and as varieties suitable for a legume crop rotation with cereals on the Tibetan plateau.

## Figures and Tables

**Table 1 animals-10-00596-t001:** Ingredients and chemical compositions (g/kg dry matter) of the experimental diet.

Items	Diet ^1^					Alfalfa Hay
	Control	C333A	CLJ1	CLJ2	CLJ3	
Ingredients (% fed basis)						
Corn straw	30.0	30.0	30.0	30.0	30.0	
Alfalfa hay	20.0					
333A stover		20.0				
Lanjian No. 1 stover			20.0			
Lanjian No. 2 stover				20.0		
Lanjian No. 3 stover					20.0	
Corn gluten feed	2.0	0.8	1.1	0.8	0.0	
Corn	30.0	30.0	30.0	30.0	30.0	
Molasses	4.0	4.0	4.0	4.0	4.0	
Soybean meal	5.0	5.0	5.0	5.0	5.8	
Cottonseed meal	5.0	5.0	5.0	5.0	5.0	
Corn gluten meal	0.7	1.9	1.6	1.9	1.9	
Limestone	0.4	0.4	0.4	0.4	0.4	
Salt	0.7	0.7	0.7	0.7	0.7	
Urea	1.2	1.2	1.2	1.2	1.2	
Mineral-vitamin premix ^2^	1.0	1.0	1.0	1.0	1.0	
F:C ratio ^3^	50.0:50.0	50.0:50.0	50.0:50.0	50.0:50.0	50.0:50.0	
Chemical composition						
Dry matter (g/kg)	889.1	900.9	891.4	898.6	887.2	889.4
Crude protein	164.4	165.3	163.4	162.2	160.4	154.4
Neutral detergent fiber	370.1	379.1	361.0	377.8	375.4	466.7
Acid detergent fiber	170.4	184.6	161.5	173.4	178.5	324.4
Ash	90.1	82.2	92.8	87.4	94.8	116.9

^1^ Control, diet containing alfalfa hay; C333A, diet containing 333A stover; CLJ1, diet containing Lanjian No. 1 stover; CLJ2, diet containing Lanjian No. 2 stover; CLJ3, diet containing Lanjian No. 3 stover; ^2^ Mineral and vitamin premix contained, g/kg: Fe 69.63, Cu 7.41, Mn 23.7, Zn 55, I 0.67, Se 0.3, Co 0.3, vitamin A 2500 IU, vitamin E 23 IU; ^3^ F:C ratio, proportion between the amounts of roughage and concentrate in diet.

**Table 2 animals-10-00596-t002:** Chemical composition (g/kg DM) and nutritive value of the stovers from four varieties of common vetch.

Item	Stover ^1^					
	C333A	CLJ1	CLJ2	CLJ3	SEM	*p*-Value
Organic matter	906.43 ^a^	883.36 ^c^	893.01 ^bc^	902.73 ^ab^	2.36	<0.001
Crude protein	117.62 ^c^	142.72 ^a^	130.60 ^b^	120.05 ^c^	2.26	<0.001
Ether extract	4.59 ^b^	5.15 ^b^	10.58 ^a^	12.05 ^a^	0.73	<0.001
Neutral detergent fiber	540.76 ^a^	454.93 ^c^	481.09 ^b^	521.40 ^a^	7.10	<0.001
Acid detergent fiber	377.36 ^a^	311.92 ^c^	336.28 ^b^	345.32 ^b^	5.04	<0.001
Acid detergent lignin	95.60 ^a^	60.80 ^c^	65.26 ^bc^	70.49 ^b^	2.70	<0.001
Hemicellulose	163.40 ^ab^	143.01 ^c^	144.81 ^bc^	176.08 ^a^	3.44	<0.001
Cellulose	281.77 ^a^	251.12 ^b^	271.02 ^a^	274.84 ^a^	3.31	0.003
Phosphorus	1.32 ^c^	3.16 ^a^	1.38 ^c^	2.69 ^b^	0.15	<0.001
Calcium	9.98 ^c^	15.38 ^a^	11.33 ^c^	13.24 ^b^	0.48	<0.001
IVOMD ^1^	496.36 ^c^	565.84 ^a^	544.72 ^ab^	517.45 ^bc^	7.24	0.001
ME	6.84 ^c^	7.83 ^a^	7.52 ^ab^	7.14 ^bc^	0.11	0.002

^a,b^ means within a row with different superscripts differ (*p* < 0.05). ^1^ IVOMD, in vitro organic matter digestibility (g/kg DM); P, phosphorus; ME, metabolizable energy (MJ/kg DM); SEM, standard error of the mean.

**Table 3 animals-10-00596-t003:** Effect of dietary treatments on the growth performance, apparent digestibility, and nitrogen balance of lambs.

Items	Diet ^1^						
	Control	C333A	CLJ1	CLJ2	CLJ3	SEM	*p*-Value
Growth performance							
Initial body weight (kg) ^2^	19.18	19.29	19.18	19.19	18.61	0.33	0.973
Final body weight (kg)	31.76	29.49	31.00	31.02	28.61	0.48	0.224
Average daily gain (g/day)	251.60 ^a^	204.00 ^bc^	236.40 ^ab^	236.50 ^ab^	200.00 ^c^	5.77	0.010
Dry matter intake (kg/day)	1.36	1.35	1.31	1.34	1.26	0.02	0.611
Feed conversion ratio ^3^	5.44 ^b^	6.71 ^a^	5.57 ^b^	5.77 ^b^	6.37 ^a^	0.11	<0.001
Apparent digestibility (g/kg)							
Dry matter	636.99	608.00	635.01	624.02	603.98	11.92	0.885
Organic matter	654.02 ^a^	565.44 ^b^	637.99 ^a^	618.02 ^ab^	573.98 ^b^	10.85	0.019
Crude protein	668.05	603.30	646.92	650.86	606.29	9.33	0.092
Neutral detergent fiber	518.52 ^a^	446.48 ^b^	510.14 ^a^	483.72 ^ab^	453.38 ^b^	9.48	0.035
Acid detergent fiber	446.60 ^a^	345.52 ^b^	426.48 ^a^	392.82 ^ab^	353.34 ^b^	12.27	0.018
Nitrogen balance (g/day)							
Intake	30.93	31.00	31.06	29.99	30.65	0.69	0.991
Fecal excretion	10.22 ^b^	12.23 ^a^	10.84 ^ab^	10.35 ^b^	11.99 ^a^	0.26	0.022
Urinary excretion	8.69	10.10	8.59	8.34	9.60	0.51	0.810
Body retention	12.03 ^a^	8.67 ^c^	11.63 ^a^	11.30 ^ab^	9.06 ^bc^	0.44	0.022
N retention (g/100 g)	38.87 ^a^	28.02 ^b^	37.74 ^a^	37.63 ^a^	29.28 ^b^	1.25	<0.001

^a,b^ Means within a row with different subscripts differ when *p* < 0.05. ^1^ Control, diet containing alfalfa hay; C333A, diet containing 333A stover; CLJ1, diet containing Lanjian No. 1 stover; CLJ2, diet containing Lanjian No. 2 stover; CLJ3, diet containing Lanjian No. 3 stover; SEM, standard error of the mean. ^2^ Initial body weight was recorded at the first day of the fattening period. ^3^ Feed conversion ratio = dry matter intake (g)/average daily gain (g).

**Table 4 animals-10-00596-t004:** Effect of dietary treatments on pH, total volatile fatty acids (VFA, mM) content, and molar proportions of individual VFA (% of total VFA) of Hu growing lambs.

Items	Diet ^1^						
	Control	C333A	CLJ1	CLJ2	CLJ3	SEM	*p* Value
pH	6.51	6.78	6.57	6.67	6.75	0.04	0.222
Total VFA	79.18 ^a^	67.84 ^b^	77.47 ^ab^	77.69 ^ab^	68.86 ^b^	1.62	0.047
Acetate	59.62 ^a^	53.50 ^b^	59.00 ^a^	58.89 ^a^	54.12 ^b^	0.78	0.008
Propionate	19.82 ^b^	25.90 ^a^	20.70 ^b^	20.71 ^b^	24.90 ^a^	0.44	<0.001
Butyrate	14.27	15.44	14.65	14.22	15.85	0.27	0.199
Valerate	2.94 ^a^	2.46 ^b^	2.82 ^a^	2.98 ^a^	2.39 ^b^	0.07	0.002
Isobutyrate	1.39	1.21	1.38	1.38	1.20	0.05	0.511
Isovalerate	1.96	1.48	1.44	1.82	1.54	0.08	0.131
Acetate:Propionate	2.65 ^a^	1.89 ^c^	2.22 ^bc^	2.56 ^ab^	2.13 ^c^	0.08	0.003

^a,b^ Means within a row with different subscripts differ when *p* < 0.05. ^1^ Control, diet containing alfalfa hay; C333A, diet containing 333A stover; CLJ1, diet containing Lanjian No. 1 stover; CLJ2, diet containing Lanjian No. 2 stover; CLJ3, diet containing Lanjian No. 3 stover; SEM, standard error of the mean.

**Table 5 animals-10-00596-t005:** Effect of replacement of alfalfa hay with different varieties of common vetch stover on methane emission of Hu growing lambs.

Items	Diet ^1^						
	Control	C333A	CLJ1	CLJ2	CLJ3	SEM	*p* Value
mmol/L ^2^	17.31 ^a^	11.89 ^b^	16.49 ^a^	16.54 ^a^	12.29 ^b^	0.53	<0.001
L/Day ^3^							
Minimum	51.93	51.72	50.53	51.43	48.75	0.70	0.610
Maximum	83.44	83.18	81.82	82.85	79.75	0.81	0.610
L/kg DOMI							
Minimum	75.21 ^b^	92.18 ^a^	79.64 ^b^	80.55 ^b^	87.28 ^a^	1.49	0.006
Maximum	124.35 ^b^	150.09 ^a^	130.41 ^b^	132.67 ^b^	143.66 ^a^	2.33	0.044
L/kg ADG							
Minimum	208.39 ^b^	257.48 ^a^	214.76 ^b^	221.64 ^b^	247.49 ^a^	4.41	<0.001
Maximum	335.34 ^b^	415.74 ^a^	348.64 ^b^	358.18 ^b^	406.11 ^a^	7.51	<0.001

^a,b^ Means within a row with different subscripts differ when *p* < 0.05. 1 ADG, average daily gain; Control, diet containing alfalfa hay; C333A, diet containing 333A stover; CLJ1, diet containing Lanjian No. 1 stover; CLJ2, diet containing Lanjian No. 2 stover; CLJ3, diet containing Lanjian No. 3 stover; DOMI, digestible organic matter intake; SEM, standard error of the mean. ^2^ Values were calculated according to the equation of Moss et al. [22]. ^3^ Values were calculated according to the equation of Ramin and Huhtanen [23].

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
