# Peer review of "Comparative Effects of Stovers of Four Varieties of Common Vetch on Growth Performance, Ruminal Fermentation, and Nutrient Digestibility of Growing Lambs"

_animals, 2020, doi:10.3390/ani10040596_

Round 1
Reviewer 1 Report
I have gone through the Manuscript (ID: animals-739748) entitled “Comparative effects of stovers of four varieties of common vetch on growth performance, ruminal fermentation, and nutrient digestibility of growing lambs”.
General comments
The manuscript is overall well written, interesting and in the aim of the journal. Hence, it can be considered for publication. However, there are some points which needs to be improved or changed.
My specific comments and suggestions for improving the paper are:
I suggest reporting the chemical composition of alfalfa. I think it can be useful to better understand and explain the results, considering that it is the only ingredient replaced with vetch stovers.
[Table 1] I suggest including “ether extract” and “acid detergent lignin” in the table 1.
[P4 L135] Why did you take the rumen fluids from only 5 animals?
Although lambs were individually fed, were 5 animals enough to provide you a good power of the analysis?
[P4 L139] Please delete “Ma et al.” or add “according to Ma et al.”
[P5 L144] Similar to above, why did you selected only 5 animals?
[P5 L160] Please delete the repetition “for 2 h”.
[P5 L167] You should better detail how the IVOMD and ME values ​​were calculated. The reference mentioned is not easy to find.
[P5 L171] Why didn’t you feed the rams with the same forage:concentrate ratio (50:50) used during the experimental trial?
[P5 L174] How did you measure the volume of GP? Please detail it in the manuscript.
[P5 L176] Specify whether the incubation runs were repeated in 3 consecutive days or differently.
[P5 L182-L183] Please explain better how these calculations were made.
[P7 L218] respectively what?
[P7 L223] Add “of” after effects.
Author Response
Response to Reviewer 1 Comments
Point 1: I suggest reporting the chemical composition of alfalfa. I think it can be useful to better understand and explain the results, considering that it is the only ingredient replaced with vetch stovers.
Response 1: We added “chemical composition of alfalfa” at Table 1 of the revised manuscript.
Point 2: Table 1 I suggest including “ether extract” and “acid detergent lignin” in the table 1.
Response 2: Thank you for your good suggestion. We are sorry to say that we did not measure the ether extract and acid detergent lignin of the experimental diet. We cannot make supplementary measurements of these parameters, due to movement restrictions related to the new coronavirus pneumonia.
Point 3: L135 Why did you take the rumen fluids from only 5 animals? Although lambs were individually fed, were 5 animals enough to provide you a good power of the analysis?
Response 3: Thanks for your very relevant comment. We agree that the greater the number of experimental animals, the more reliable the data analysis will be. However, there is precedent in other published papers for the number of test sheep for ruminal fermentation parameters and nutrient digestibility to be as low as 4 (Lv et al., 2019; Saro et al., 2019) and 5 (Li et al., 2012).
Lv, X.K., Cui, K., Qi, M.L., Wang, S.Q., D, Q.Y., Zhang, N.F. Ruminal microbiota and fermentation in response to dietary protein and energy levels in weaned lambs. Animals 2020, 10(1), 109.
Saro, C., Mateo, J., Andrés, S., Mateo, I., Ranilla, M.J., López, S., Martín, A., Giráldez, F.J. Replacing soybean meal with urea in diets for heavy fattening lambs: effects on growth, metabolic profile and meat quality. Animals 2019, 9(11), 974.
Li, L., Davis, J., Nolan, J., Hegarty, R. An initial investigation on rumen fermentation pattern and methane emission of sheep offered diets containing urea or nitrate as the nitrogen source. Anim. Prod. Sci. 2012, 52, 653-658.
Again, in the present paper, the statistical analysis indicates that response differences between treatments were established with this number of animals, and from a logistical perspective, 25 metabolic crates with animals (5 x 5 treatments) is a seriously large experiment.
Point 4: L139 Please delete “Ma et al.” or add “according to Ma et al.”
Response 4: We deleted “Ma et al.” at Line 140 of the revised manuscript.
Point 5: L144 Similar to above, why did you selected only 5 animals?
Please see response to point 3 above.
Point 6: L160 Please delete the repetition “for 2 h”.
Response 6: We deleted “for 2 h” at Line 159 of the revised manuscript.
Point 7: L167 You should better detail how the IVOMD and ME values were calculated. The reference mentioned is not easy to find.
Response 7: As suggested by the reviewer, we added the sentence “as IVOMD = 148.8 + 8.89 GP + 0.45 CP + 0.0651 ASH; ME = 2.20 + 0.136 GP + 0.0057 CP + 0.000029 EE2, where GP is the net gas production after 24 h of incubation (mL/200mg DM), ASH is the ash content (g/kg DM), CP is the crude protein content (g/kg DM), and EE is the ether extract content (g/kg DM).” at lines 177-180 of the revised manuscript.
Point 8: L171 Why didn’t you feed the rams with the same forage:concentrate ratio (50:50) used during the experimental trial?
Response 8: Thanks for your insightful comment. When the experiment was designed, we only thought of obtaining rumen fluid as the key solution for measuring gas production parameters, so we did not consider the use of diet with the same forage:concentrate ratio. This comment is valuable and very helpful for our future research.
Point 9: How did you measure the volume of GP? Please detail it in the manuscript.
Response 9: As suggested by the reviewer, we added the sentence “The syringes were held vertically at eye level and a gas reading of each syringe was manually recorded.” at lines 175-176 of the revised manuscript.
Point 10: L176 Specify whether the incubation runs were repeated in 3 consecutive days or differently.
Response 10: As suggested by the reviewer, we changed “repeated three times.“ to “ three times on different days.” at line 181 of the revised manuscript.
Point 11: L182-L183 Please explain better how these calculations were made.
Response 11: A clarification was added at lines 187-189 in the revised manuscript: “(negative values were used to estimate the minimum value and positive values were used to estimate the maximum value for methane emission)
Point 12: L218 respectively what?
Response 12: We deleted “respectively” at Line 226 of the revised manuscript.
Point 13: L223 Add “of” after effects.
Response 13: We added “of” at Line 231 of the revised manuscript.
Additional minor revisions were made to the manuscript using track changes to improve English and readability, based on suggestions from our co-author whose native language is English.
Special thanks for your good comments. We hope that the corrections made will meet with approval.

Reviewer 2 Report
This manuscript is very well written, all sections are clear. There only are a few minor comments.
L124-126 Were the lambs weighed only one at the start and at the end of the fattening period? To adjust for rumen fill, the animals should preferably be weighed on two consecutive days at start and at the end and the mean of the BW from the 2 days should be used.
L139 Delete Ma et al.
L150 Was urine collected during the last 3 days as you did collection of the faeces. Please add.
Table 3 Describe feed conversion ratio; feed, g/ADG, g
Footnot: Replace raw with row.
L297 Delete “that fed”
L305 Write ADL contents of these stovers
L332-333 Clarify the sentence!
L356-357 CLJ1 did not differ in CP, IVOMD and ME from CLJ2. Reword.
357 Replace ME levels with ME contents.
Author Response
Response to Reviewer 2 Comments
Point 1: L124-126 Were the lambs weighed only one at the start and at the end of the fattening period? To adjust for rumen fill, the animals should preferably be weighed on two consecutive days at start and at the end and the mean of the BW from the 2 days should be used.
Response 1: In fact, the initial and final weights were both two-day averages. Body weight and ADG were calculated using this method. The purpose of this previous expression is to make it easy to read and understand. And this is now clarified at lines 124-125 of the revised manuscript.
Point 2: L139 Delete Ma et al.
Response 2: As suggested by the reviewer, we deleted “Ma et al.” at line 140 of the revised manuscript.
.
Point 3: L150 Was urine collected during the last 3 days as you did collection of the faeces. Please add.
Response 1: Yes, it was. We have revised this line to read “The digestibility trial lasted for 7 days with a four day adaptation period, which was followed by the total collection of feces and urine for three consecutive days.”at lines 146-147 of the revised manuscript.
Point 4: Table 3 Describe feed conversion ratio; feed, g/ADG, g; Footnot: Replace raw with row.
Response 4: As suggested by the reviewer, we added “Feed conversion ratio = dry matter intake (g) / average daily gain (g).” at lines 223-224 of the revised manuscript.
We changed “raw“ to “row” at Tables 1-5 of the revised manuscript.
Point 5: L297 Delete “that fed”
Response 5: We changed “that fed” to “animals fed” at line 302 of the revised manuscript, which the coauthor who is a native English speaker advised us would be more correct.
Point 6: L305 Write ADL contents of these stovers
Response 6: As suggested by the reviewer, we changed “ADL contents these stovers“ to “ADL contents of these stovers” at line 310 of the revised manuscript. We also changed “lesser” to “lower” in the same line
Point 7: L332-333 Clarify the sentence!
Response 7: As suggested by the reviewer, we changed “Variable responses of ruminal VFA among the dietary treatments differences between the diets tested for apparent digestibility of nutrients.“ to “Variable responses of ruminal VFA among the dietary treatments reflected differences between the diets tested in apparent digestibility of nutrients.” at lines 337-338 of the revised manuscript.
Point 8: L356-357 CLJ1 did not differ in CP, IVOMD and ME from CLJ2. Reword.
Response 1: As suggested by the reviewer, we added “with the exception of the IVOMD, ME and ADL contents from CLJ2 stover.“ at lines 363-364 of the revised manuscript.
Point 9: L357 Replace ME levels with ME contents.
Response 9: As suggested by the reviewer, we changed “ME levels“ to “ME contents” at line 363 of the revised manuscript.
Additional minor revisions were made to the manuscript using track changes to improve English and readability, based on suggestions from our co-author whose native language is English.
Special thanks for your good comments. We hope that the corrections made will meet with approval.

Round 2
Reviewer 1 Report
Thank you very much for the revised version. I appreciate changes that have been made, especially those I suggested. In my opinion, now it is acceptable for the publication in the present form
Best regards